# A method for multiplexed full-length single-molecule sequencing of the human mitochondrial genome

Ieva Keraite [1], Philipp Becker [1,9], Davide Canevazzi [1], Cristina Frias-López[1], Marc Dabad [1], Raúl Tonda-Hernandez[1], Ida Paramonov[1], Matthew John Ingham[1], Isabelle Brun-Heath [2,3], Jordi Leno[4,5], Anna Abulí[4,5], Elena Garcia-Arumí[4,6,7], Simon Charles Heath [1,8], Marta Gut [1,8] ✉ & Ivo Glynne Gut [1,8] ✉

Methods to reconstruct the mitochondrial DNA (mtDNA) sequence using short-read sequencing come with an inherent bias due to amplification and mapping. They can fail to determine the phase of variants, to capture multiple deletions and to cover the mitochondrial genome evenly. Here we describe a method to target, multiplex and sequence at high coverage full-length human mitochondrial genomes as native single-molecules, utilizing the RNA-guided DNA endonuclease Cas9. Combining Cas9 induced breaks, that define the mtDNA beginning and end of the sequencing reads, as barcodes, we achieve high demultiplexing specificity and delineation of the full-length of the mtDNA, regardless of the structural variant pattern. The long-read sequencing data is analysed with a pipeline where our custom-developed software, baldur, efficiently detects single nucleotide heteroplasmy to below 1%, physically determines phase and can accurately disentangle complex deletions. Our workflow is a tool for studying mtDNA variation and will accelerate mitochondrial research.

Mitochondria are organelles found in most eukaryotic cells containing several copies of a circular genome encoding for vestigial functions of what once used to be a free-living organism. Within the human population, heteroplasmy is common[1–3] and can have substantial impact on human health depending on the variant position, frequency and other endogenous or exogenous factors[4–9]. Accurate detection and quantification of such variants is important for diagnosis, genetic counselling and treatment of mtDNA diseases.

Early analytical approaches[10–18] were revolutionised by massively parallel sequencing to detect and quantify mtDNA variants[19–21]. However, short-read sequencing methods include pre-amplification by PCR, which itself adds polymerase errors and amplification related

bias[16,22–27], and are limited by the read length, which hampers comprehensive mtDNA analysis to identify large-scale rearrangements and to phase alternative variants[28–30]. To overcome such limitations, Oxford Nanopore Technologies (ONT) and Pacific Biosciences (PacBio) with long-reads are capable of spanning most of the complex regions of the genome. One of the recent breakthroughs was enrichment to favour adaptor ligation at the cut-sites of RNA-guided Cas9[31,32]. The method has been combined with the ONT[31,32] and PacBio[33] long-read sequencing platforms.

The bioinformatics analysis of ONT sequencing data and of mitochondrial sequences both pose particular challenges. The long-read length and relatively high error rate of ONT sequences make

[1]CNAG-CRG, Centre for Genomic Regulation (CRG), The Barcelona Institute of Science and Technology (BIST), Barcelona, Spain. [2]Institute for Research in Biomedicine (IRB Barcelona) - The Barcelona Institute of Science and Technology (BIST), Barcelona, Spain. [3]Joint IRB-BSC Program in Computational Biology, Barcelona, Spain. [4]Department of Clinical and Molecular Genetics and Rare Disease, Hospital Universitari Vall d'Hebron, Barcelona, Spain. [5]Medicine Genetics Group, VHIR, Hospital Universitari Vall d'Hebron, Barcelona, Spain. [6]Research Group on Neuromuscular and Mitochondrial Disorders, VHIR, Hospital Universitari Vall d'Hebron, Barcelona, Spain. [7]Centro de Investigación Biomédica en Red de Enfermedades Raras (CIBERER), Instituto de Salud Carlos III, Barcelona, Spain. [8]Universitat Pompeu Fabra, Barcelona, Spain. [9]Present address: Qiagen, Hilden, Germany. ✉e-mail: marta.gut@cnag.crg.eu; ivo.gut@cnag.crg.eu

accurate detection of SNVs and short indels very challenging, and specialised tools have been developed for this[34,35]. The analysis of mitochondrial sequences is also challenging for different reasons: (a) the genome is circular, which makes mapping of reads across the reference origin problematic, particularly with long-reads where a large proportion of the reads will span the origin, and (b) mtDNA variant calling is closer to somatic than germline variant calling, as any variant frequency is possible and variants can arise independently in different mitochondria, potentially leading to several sub-clones. In addition, nuclear mitochondrial sequences (NUMTs)[36,37] can interfere in the bioinformatics analysis due to sequence similarity[38–41]. While variant calling pipelines for mitochondrial sequences do exist (GATK has a mitochondrial specific pipeline: https://github.com/gatk-workflows/gatk4-mitochondria-pipeline), they are not adapted to the peculiarities of ONT sequence data.

Here we present a method to target the mitochondrial genome selectively, and sequence the entire native molecule in a single read starting from a Cas9 cut-site, which serves, at the same time, as the beginning and end delineator of the full-length reads and as a multiplexing barcode to facilitate the workflow and decrease the cost per sample. The combination of our method with the ONT Q20+ ligation sequencing kit (SQK-LSK112, colloquially referred to as Kit12 or Q20+ chemistry) and the R10.4 flow cells[42], improves the overall read accuracy and the accuracy of homopolymer sequencing. Our laboratory workflow benefits from a custom-developed informatics pipeline, which allows reliable detection of low-level heteroplasmies and disentangles complex deletion patterns while having modest computing requirements allowing it to be run on an average workstation. We also demonstrate that the Cas9-mtDNA-enrichment and our analytical workflow is suitable, with minor changes in the laboratory procedures, for low quality genomic DNA (gDNA) samples, extending the use for non-invasive sampling. Translating this method into clinical research elevates our capacity to characterise the role of mtDNA in health and disease, opens a new avenue for population studies and can easily extend into other eukaryotes' mitochondrial research.

## Results
### Preparing the Cas9-mtDNA-enrichment sequencing library
We adapted and further developed laboratory procedures of Cas9-mtDNA-enrichment to allow highly selective single-molecule sequencing of mtDNA from a human gDNA sample. The method is amenable for multiplexing (Fig. 1) without the need for additional steps related to barcoding and with minor changes suitable for gDNA samples with low integrity. The first step of mtDNA enrichment is using Exonuclease V digestion. This step is optional and is highly recommended for gDNA samples with high integrity to pre-enrich the circular mtDNA. The Exonuclease V digestion eliminates linear gDNA and this way also minimises the presence of the NUMTs. For samples with low integrity this step is omitted as Exonuclease V digestion of linearised mtDNA due to degradation is not desired. To decide about including the Exonuclease V digestion, we established gDNA integrity quality control measures, such as high-resolution DNA sizing and long-range PCR (lrPCR), the latter also helps to confirm the presence of large structural variants in the sample. The second step of the mtDNA enrichment relies on the dephosphorylation of all available DNA 5′-ends in the gDNA sample, which is crucial to prevent ligation of nanopore sequencing adaptors to all non-targeted DNA fragments. After the nuclear DNA dephosphorylation step, a gDNA sample is split into aliquots and mtDNA in each aliquot is cleaved sequence-specifically by dual-RNA-guided Cas9 endonuclease (Fig. 1). This third mtDNA enrichment step selectively induces one double-strand break at a specific position in the mtDNA, which is used as a barcode so that multiple samples and their aliquots, cleaved at different positions, can be joined for library preparation and sequenced on one sequencing flow cell without a need for additional indexes. The Cas9 cut-barcode also determines the full-length of the mtDNA as the molecule starts and ends with the same guide-directed break. In our setting, we used two guide RNAs (gRNAs) per sample and to multiplex up to four samples in a GridIon flow cell (Supplementary Note 1 and Supplementary Table 1). This approach was modified to a triplet of gRNAs for comprehensive characterisation of a clinical sample with multiple mtDNA deletions. After the Cas9 cleavage we implemented the last the mtDNA enrichment step. Aliquots are treated with Proteinase K to remove the Cas9 protein bound to the mtDNA, pooled, and the ONT sequencing adaptors are ligated to the phosphorylated Cas9 cleavage sites. Method optimisation is provided in the Supplementary Information (Supplementary Note 1 and 2, Supplementary Fig. 1, Supplementary Table 1 and Supplementary Data 1–5). For data analysis a dedicated informatics pipeline was developed (Fig. 2).

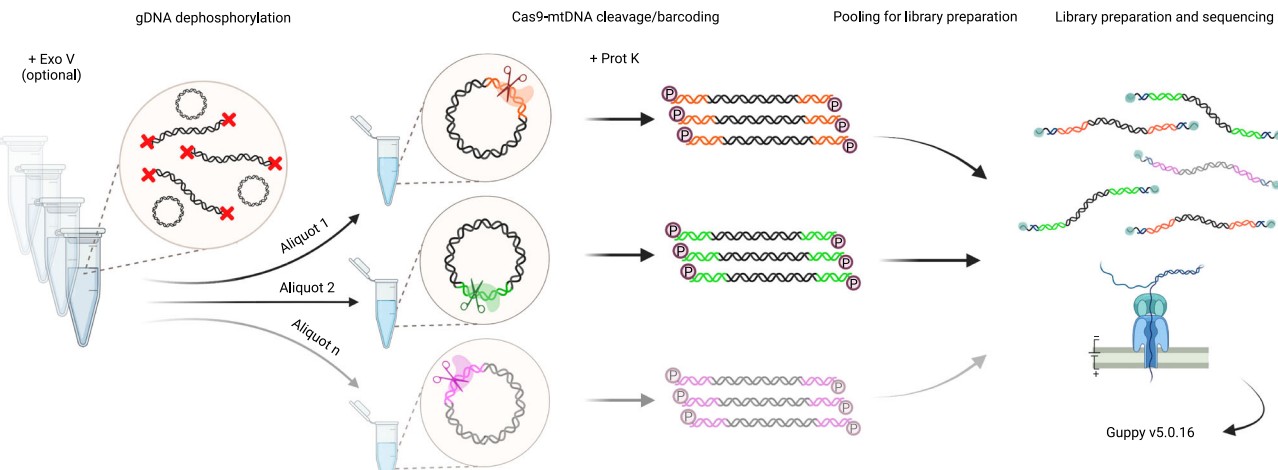

**Fig. 1 | Cas9-mtDNA-enrichment, barcoding, pooling and demultiplexing approach for long-read sequencing.** A schematic overview of full-length mtDNA targeting with selected dual-guide Cas9 cut-sites. After optional treatment of gDNA with Exonuclease V (Exo V) and dephosphorylation, each sample is split into two or more aliquots for dual-guide targeted cleavage. Each cut-site serves downstream as a barcode in the analysis pipeline. The circular mtDNA molecules are opened, Cas9 is removed by Proteinase K (Prot K) digestion, followed by mtDNA dA-tailing and pooling of all of the aliquots. ONT library is prepared from the pooled samples, sequenced on a nanopore flow cell, followed by basecalling (Guppy v5.0.16). Figure created with BioRender.com.

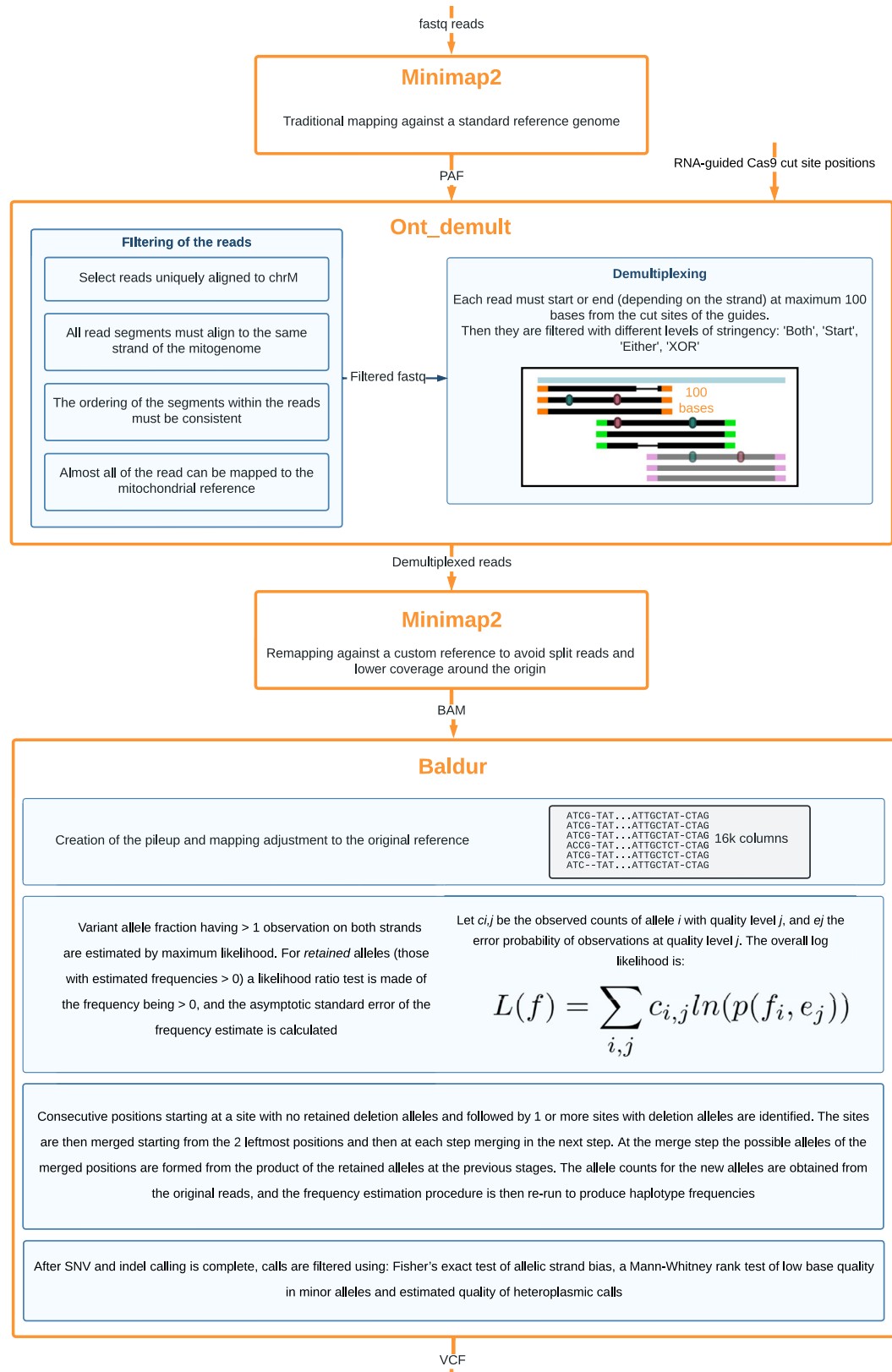

**Fig. 2 | Flowchart detailing informatics analysis pipeline steps for Cas9-mtDNA-enrichment sequencing.** After sequencing and basecalling, FASTQ reads are mapped against the reference GRCh38. Afterwards the reads are filtered and demultiplexed. The second mapping against a custom reference is critical for avoiding split reads. Then the baldur software does the variant calling for each demultiplexed sample. Resulting VCFs are used for downstream analysis for the annotation of the variants. Created in Lucidchart (www.lucidchart.com).

## Demultiplexing

Demultiplexing of Cas9-mtDNA-enriched long-reads (Supplementary Note 3) was performed by aligning the reads using minimap2 to the human whole genome reference GRCh38 to produce a PAF alignment file (Fig. 2). Aligning mitochondrial reads can be problematic since alignment tools assume linear reference sequences, while the mitochondrial genome is circular. This can cause low levels of mapping of short sequence reads that span the transition between the end and start of the mitochondrial chromosome reference. With long-reads, however, this problem does not occur, so a normal alignment procedure could be used for the demultiplexing step. However, a large fraction of the reads produced split or supplementary alignments where different segments of the read align to separate locations. This is expected for mitochondrial reads, as any read that spans the end of the reference sequence will map as two segments, one ending at the end of the reference and the other starting at the beginning of the reference.

The PAF alignment records were used to select reads that uniquely aligned to the mitochondrial genome and, from these, to further select reads that started and/or ended near a cut-site (Fig. 2). The selected reads were also filtered to ensure that (a) all read segments aligned to the same strand of the mitochondrial genome; (b) the ordering of the segments within the read and along the chromosome were consistent (allowing for the circularity of the chromosome) and (c) almost all of the read could be successfully mapped to the mitochondrial reference sequence. Minimap2 can produce PAF (minimap2 specific) or a standard SAM output. PAF output was used for demultiplexing, as it is a simpler format than SAM that made performing the previously described filtering easier.

Four different selection strategies for reads based on the mapping to cut-sites were developed with different levels of stringency. The most stringent strategy, 'Both', requires that a read must both start and end near the same cut-site. This has the effect of only selecting full-length reads (those that span the entire mitochondrial genome). The advantage of this strategy is that the risk of mis-assigning a read during demultiplexing is very low, the coverage across the mitochondrial reference is completely even in the absence of deletions, and the full-length reads do not jeopardise native read ratios due to length bias. If there are not enough full-length reads (e.g., if the gDNA sample has low integrity) then less stringent strategies can be used. The 'Start' strategy only requires that a read starts near a cut-site, while the 'Either' strategy will select a read if it starts or ends near a cut-site. The less stringent strategies produce higher coverage but have the drawbacks of having a marginally increased risk of mis-assignment of reads during the demultiplexing process, and also have uneven coverage across the mitochondrial genome. A fourth selection strategy was also tested purely for benchmarking and was designed to simulate the situation of dealing with highly degraded DNA when no full-length reads can be obtained. With this strategy, 'XOR', reads are selected either if the start or the end of the read matches a cut-site, but not both. In this way, full-length reads are not selected with this strategy.

For all selection strategies, the matching of a read to a cut-site was performed in the same way: the start (end) of a read matched a cut-site if it lay not more than 100 bp after (before) the cut-site, taking the strand of the read into account when determining the direction (Fig. 2). For example, a full-length read, selected with the 'Both' strategy, will either start at or just after a cut-site and finish at or just before the same cut-site if on the positive strand, or start at or just before a cut-site and finish at or just after the same cut-site if on the negative strand.

## Remapping

The result of the demultiplexing step was a separate FASTQ file per cut-site, with each file only containing the reads that had been allocated to that cut-site given the chosen selection strategy (described above). Each FASTQ file, therefore, only contains reads that map uniquely to the mitochondrial genome, with a start and/or end point (depending on the selection strategy) that corresponds to the respective cut-site (Supplementary Note 4), and whose mapped length (i.e., the distance between the start and end point on the genome) is ≤ the mitochondrial genome length.

Mapping of short sequence reads (i.e., read lengths of 250 bp or less) to a circular genome can cause difficulties for reads that span the reference origin of the genome, resulting in low mapping efficiencies for such reads. With the longer reads from the nanopore sequencing, the mapping of reads spanning the origin is not a problem in itself, as the reads can be mapped in multiple segments to give split alignments. However, it is common that the aligner will not be able to precisely determine the split point between segments resulting in the bases around the origin having lower coverage than the rest of the genome. To avoid this, we performed a remapping step using a custom reference sequence for each cut-site.

The custom reference sequences all consist of 2 tandem repeats of the mitochondrial genome with a small number $d$ of bases added as padding to each end of the new reference giving a total length of $2 \times (16569 + d)$ bp (Methods section and Supplementary Fig. 2). As the mitochondrial genome is duplicated in this scheme, each cut-site appears twice, once in each repeat. For each cut-site, a custom reference is made so that the bases before the location of the cut-site in repeat 1 and after the location of the cut-site in repeat 2 are masked. In this way both full length and partial length reads that either start or end at the cut-site will map uniquely to the custom reference. This scheme to generate the custom references allows us to conserve the same coordinate system for all the cut-sites and therefore to merge the alignment files obtained with different gRNA for a given sample.

Remapping was performed using minimap2, this time generating SAM output, which was converted to BAM for the variant calling stage described below (Fig. 2). The different alignment files belonging to each sample (as each sample had reads from multiple cut-sites) were merged using samtools before the subsequent steps, but the alignment records were left in read order (i.e., the BAM was not sorted by coordinate).

## Sequence variant calling, annotation and reporting

SNV and short indel calls were made using the programme baldur developed in-house (Fig. 2 and Supplementary Note 5). The accuracy of the calling was checked by comparing to the calls produced by the GATK mitochondrial calling pipeline on an Illumina dataset from the same samples (Supplementary Fig. 3). The strategy for variant calling used by baldur is described below.

1. The BAM for a sample was read in and a sequence pileup was generated in memory storing information on the strand, bases, qualities and insertions. The position x of each base was adjusted using the function $x' = (x-d) \mod 16569$ to map back to the original reference positions.
2. At each position, the number of observations of each allele on each strand was recorded, and the variant allele fraction (VAF) of alleles having >1 observations on both strands were estimated by maximum likelihood (described in detail in Supplementary Note 5). For retained alleles (those with estimated VAF > 0), a likelihood ratio test was made of the frequency being >0, and the asymptotic standard error of the frequency estimate was calculated.
3. Long regions of consecutive sites having a retained deletion allele were identified. All long deletion alleles in the region were extracted, and alleles with similar start points and lengths were merged. After this step, the single site frequencies were re-estimated for the positions covered by the deletion(s), not taking into account the reads carrying the long deletion alleles.
4. Consecutive positions starting at a site with no retained deletion alleles and followed by 1 or more sites with deletion alleles were identified. The sites were then merged starting from the 2 left-most positions and then at each step merging in the next step. At

the merge step the possible alleles of the merged positions were formed from the product of the retained alleles at the previous stages. The allele counts for the new alleles were obtained from the original reads, and the VAF estimation procedure was then re-run to produce haplotype frequencies.

5. After SNV and indel calling is complete, calls were filtered using (a) a Fisher's exact test of allelic strand bias; (b) a Mann–Whitney rank test of low base quality in minor alleles; (c) Estimated quality of heteroplasmic calls. VCF output was then produced for called variants.

After variant calling, VCF files were passed to the annotation pipeline (Fig. 2). Variant annotations from the regularly updated resources dedicated to mtDNA were added to the VCF files: (1) disease-associated mutations and polymorphisms along with the GenBank frequencies from MITOMAP;[43] (2) population frequencies for mtDNA variants from gnomAD;[44] (3) in silico pathogenicity prediction scores from MitoTIP[45]. In addition, mitochondrial haplogroup assignment was conducted for each sample with Haplocheck[46]. The complete variant annotation result output is summarised in the clinical samples variant annotation table (Supplementary Data 6).

SNV phasing was investigated from the pileup of all reads using the programme baldur with the command line option --view. This programme produces a text file with 16,569 columns, one for each position in the mitochondrial genome, and one row per read showing sequence bases, deletions and skips. For this file, the phasing of any set of positions (e.g., a set of heteroplasmic positions) can be extracted and the frequencies obtained (Supplementary Fig. 4). Note that this produces phasing only for the selected SNVs. This allows phasing of distant SNVs to be obtained simply without encountering problems due to sequence variability (whether true or due to sequencing errors) in any intervening bases.

### Improving average divergence per base

Until recently, nanopore sequencing was associated with a high error rate of up to 10%. The Q20+ chemistry together with the improved double-reader head R10.4 pore flow cells were introduced to ameliorate the single-molecule read accuracy, especially in complex genomic regions with homopolymers. Originally, we used the ligation sequencing kit SQK-LSK110 (Kit 10), Supplementary Data 3. Changing to the Q20+ chemistry resulted in eminent improvement across all types of GridIon flow cells currently available from ONT (Supplementary Data 4). The average divergence per base measured was $1.07 \pm 0.03\%$ on R9.4.1 and $1.05 \pm 0.04\%$ on R10.3 ($P = 0.3404$), while R10.4 demonstrated a significant improvement in comparison to R9.4.1 flow cell, $1.00 \pm 0.04\%$ vs. $1.08 \pm 0.03\%$ ($P = 0.0043$), respectively.

### Cas9-mtDNA-enrichment analysis identifies heteroplasmy in pathogenic SNVs

We applied our enrichment method, without the Exonuclease V treatment, on a set of 14 clinical samples with confirmed heteroplasmy of pathogenic SNVs provided by a diagnostic laboratory (Table 1). The gDNA samples originated from different clinical specimens (e.g., blood, muscle, urine, oral mucosa; Supplementary Table 2) and consequently the extracted gDNA had variable quantity and integrity. The mtDNA genome coverage using only the mtDNA full-length reads was between ×33 and ×2335, where the lowest coverage was in samples from oral mucosa and urine, consistent with the DNA degradation level determined from the DNA quality control (Supplementary Note 6 and Supplementary Fig. 5). We ran each low integrity sample in a separate sequencing flow cell, and we included in the analysis the full-length reads ('Both' in Table 1) and the reads that started at the specific Cas9 introduced cut-site barcode but that were not exclusively full-length ('Start' in Table 1). This strategy helped to increase the coverage 4-fold for these degraded samples and increased confidence of variant calling. The blood extracted DNA samples were less degraded, thus this strategy provided a less significant boost, mostly <2-fold, also these samples could be confidently multiplexed (data is provided in Supplementary Data 1). We could confirm all heteroplasmies reported by the diagnostic laboratory with consistent frequency estimates (Table 1) ranging from <0.2% to 100%. No large deletions were observed by our method or the lrPCR quality control (Supplementary Note 7, Supplementary Fig. 6 – Samples (2)). In addition, non-pathogenic

**Table 1 | Pathogenic SNV identification and heteroplasmy determination in the confirmed clinical samples with mtDNA alterations**

| Sample ID | Material source | Gene | Variant | Diagnostic laboratory results | | Cas9-mtDNA-enrichment ONT sequencing | | | |
| --- | --- | --- | --- | --- | --- | --- | --- | --- | --- |
| | | | | Heteroplasmy % | Depth | Heteroplasmy % | | Depth | |
| | | | | | | 'Both' | 'Start' | 'Both' | 'Start' |
| AW6491 | Oral mucosa | MT-TL1 | m.3243A>G | 6.5 | 82,304 | –[b] | 4.8 | 33 | 136 |
| AW6492 | Oral mucosa | MT-TL1 | m.3243A>G | 21.1 | 126,785 | 14.9 | 18.8 | 60 | 268 |
| AW6494 | Urine | MT-TL1 | m.3243A>G | 6.3 | 91,106 | 7.5 | 6.9 | 99 | 370 |
| AW6501 | Blood | MT-TL1 | m.3243A>G | 2.26 | 118,112 | 1.4 | 1.3 | 614 | 1002 |
| AW6500 | Blood | MT-TL1 | m.3243A>G | 0.7 | 94,171 | 0.2[c] | 0.3[c] | 1318 | 1843 |
| AW6495 | Blood | MT-ND5 | m.12781A>G | 15.9 | 26,056 | 15.8 | 17.5 | 439 | 804 |
| AW6496 | Blood | MT-ATP6 | m.9185T>C | 98.6 | 21,634 | 100 | 99.4 | 106 | 339 |
| AW6497 | Blood | MT-ND1 | m.4171C>A | 25.1 | 28,142 | 25.7 | 24.9 | 605 | 748 |
| AW6498 | Blood | MT-RNR1 | m.1555A>G | 100 | – | –[a] | 100 | 100 | 2335 | 2942 |
| AW6499 | Blood | MT-RNR1 | m.1555A>G | 100 | – | –[a] | 100 | 99.9 | 1307 | 1771 |
| AW6502 | Blood | MT-TL1 | m.3243A>G | 3.63 | 99,649 | 3.9 | 3.3 | 456 | 747 |
| AW6503 | Blood | MT-TL1 | m.3243A>G | 17.3 | 73,331 | 19.1 | 18.5 | 1443 | 1944 |
| AW6505 | Blood | MT-TW | m.5541C>T | 37.2 | 18,586 | 39.2 | 39.0 | 247 | 366 |
| AW6493 | Muscle | MT-TK | m.8344A>G | 89.7 | 4776 | 89.1 | 89.7 | 745 | 4905 |

The pathogenic variants identified by the Cas9-mtDNA-enrichment nanopore sequencing were compared with the diagnostic laboratory results using Illumina sequencing. The low integrity gDNA samples display low coverage in the 'Both' bioinformatics analysis, while the 'Start' strategy allows boosting coverage. The determined heteroplasmies in all clinical samples are concordant with the diagnostic laboratory results.
[a]Sanger sequencing.
[b]The variant was only seen on one strand due to the low number of full-length reads.
[c]Flagged as low frequency in VCF.

polymorphic sites were identified in all samples mostly in homoplasmy or high-level heteroplasmy (>99%) along with variants of unknown significance (Supplementary Data 6). Assessing variant frequency within the general population and specific haplogroups is helpful when evaluating the pathogenicity of such variants[47].

### Cas9-mtDNA-enrichment analysis identifies multiple deletions

To scrutinise our method to detect multiple mtDNA deletions, we used a gDNA (AW6506) sample extracted from a muscle biopsy of an entangled clinical case known to have complex mtDNA SVs. We analysed the gDNA sample without prior information about the positions of potential deletions. In this case, the diagnostic laboratory had been able to identify the exact position only for a smaller deletion, and an approximate position of a larger deletion, using lrPCR and Illumina sequencing. The copy number of these structural variants was not quantified precisely due to high PCR biases.

In our quality control by lrPCR, we observed two bands representing two mtDNA populations of ~13.5 kb and ~3.5 kb in this sample. Wild-type 16.5 kb mtDNA was not visible on the gel, due to the amplification bias against larger fragments (Supplementary Fig. 6 – Sample (3)). The presence of the wild-type mtDNA molecule and both deletions was confirmed by Illumina and ONT sequencing of the lrPCR product, however, the sequencing results were heavily disproportioned by the amplification bias (Fig. 3a, b). Considering the limited gDNA sample availability and the complexity of the analysis, we used three gRNAs to identify the different SVs and processed this sample on a GridIon sequencing flow cell on its own. We selected three gRNAs, at positions m.3127 (mt3), m.5142 (mt5) and m.11239 (mt11), to span the mitochondrial genome (Fig. 3c–e). Using just the full length reads (using the 'Both' strategy) deriving from the mt3 cut-site, baldur successfully detected two large deletions present in the sample. A large deletion m.3264_16070del, reported previously by others[48], was detected at a low-level heteroplasmy (7.6%) covering a segment encoding 13 polypeptides crucial for the oxidative phosphorylation. A small deletion m.10751_14129del containing 3 essential polypeptides[49] was detected in 84.1% of reads. The wild-type mtDNA was present in 8.3% of reads. By comparing the nanopore sequencing results from the lrPCR amplicons and Cas9-mtDNA-enrichment, we exemplify the benefit of the native mtDNA sequencing. We observed the significant bias of preferential short molecule build-up resulting in inaccurate estimates of the three mtDNA populations in the lrPCR nanopore sequencing (Fig. 3f).

## Discussion

Here we present an efficient and selective targeting method for the human mitochondrial genome delineating its native full-length sequence status using dual-RNA-guided Cas9 nuclease and nanopore long-read sequencing. While using the Exonuclease V and Proteinase K improves the mtDNA enrichment, the different cut-sites are used to multiplex and pool several samples for library preparation. The method achieves high and even coverage along the mitochondrial genome after sequencing the pool on a single GridIon flow cell. The method was scrutinised for the effectiveness of multiplexing and precise targeting of full-length mtDNA molecules followed by sensitive detection of SNVs or disentangling complex multiple deletions in 15 clinical samples. Our study revealed evident benefits of applying the Cas9-mtDNA-enrichment protocol and nanopore native sequencing, exemplified by the distorted lrPCR amplicon representation. While variant calling pipelines exist both for ONT data and for mitochondrial genomes, there is no current pipeline that is designed to do both. By developing a custom analysis pipeline, it was possible to take advantage of the unorthodox biological characteristics of the mitochondrial genome and the particularities of long-reads from the nanopore sequencing to produce a computationally efficient and accurate pipeline for the demultiplexing, variant calling of SNVs,

small indels and long deletions, filtering and annotation, which can be run on a laptop. Testing the pipeline against an Illumina dataset analysed with the GATK mitochondrial calling pipeline showed complete concordance between the ONT results using the R10.4 flow cells with Q20+ chemistry and Illumina results for all variants with VAF ≥ 5%. Treating the Illumina results as the truth, the pipeline attains a sensitivity of 88% and a precision of 95% to detect heteroplasmic variants with VAF ≥ 0.5%, and allows accurate estimation of SNVs phasing frequencies. In addition, our system allows for complex deletions to be detected and characterised with accurate estimation of breakpoints and frequencies. It is important to note, that while the experimental system that we present used the Q20+ chemistry paired with the R10 flow cells with increased sequencing accuracy, in particular in homopolymer regions, when compared to the earlier chemistry, we obtained comparable levels of sensitivity and precision with the old, pre-Q20+ chemistry paired with a R9.4.1 flow cell (Supplementary Data 4 and 5). This indicates that the appropriate analysis and the increased coverage, given by the older chemistry, can compensate for the lower sequencing accuracy (Supplementary Data 3).

Before, the detection of the mitochondrial sequence variants was a multistep process involving several molecular methods[29]. Designed to enrich the full-length native mitochondrial genome in a single test our method offers a cost-effective human mtDNA sequence analysis with high coverage for accurate detection of SNVs, to define accurately single or multiple deletions, and to reconstruct haplotype lineages. Together with its analytical pipeline it is a promising tool for translational research to fill a void in current approaches to molecular diagnosis of mtDNA-related disorders and provides leverage to mtDNA population studies, including the confounding biases related to NUMTs. Cas9-mtDNA-enrichment could potentially be used for the investigation of somatic mutation events in the mitochondrial genome. Directly sequencing the native DNA that is captured by the Cas9-mtDNA-enrichment also opens the door to investigate any DNA modifications the mtDNA might harbour. The protocol will likely enable extending the subject beyond human mtDNA to animal species where mtDNA is a relatively conserved molecule that makes it an ideal genetic marker for conservation genetics, population genetics and molecular phylogenetics[50] to learn about the tree of life through the two billion years of symbiosis fundamental to the eukaryotes' life.

## Methods

### Ethical statement

A set of 15 clinical samples was previously collected by the Department of Clinical and Molecular Genetics and Rare Disease at the University Hospital Vall d´Hebron within a diagnostic workflow for subjects presenting with symptoms related to mitochondrial disease. The diagnostic procedures for these patients followed the internal protocols of the University Hospital Vall d´Hebron. The 15 samples included in the present work were selected according to the Department of Clinical and Molecular Genetics and Rare Disease at the University Hospital Vall d´Hebron mtDNA status results. All subjects gave informed consent approved by the bioethics committee (Clinical Research Ethics Committee (CEIC) of the Vall d'Hebron Hospital; No CI/0602/2013) and in accordance with the Declaration of Helsinki. There was no compensation for participating in the study.

### Statistics and reproducibility

The experimental design included both human cell lines and clinical samples. The cell lines allowed multiple experiments to be conducted on the same biological material, permitting the reproducibility and accuracy of the variant calling from ONT sequencing data to be assessed by comparing across multiple ONT sequencing experiments, and by comparing estimates from ONT and Illumina sequencing

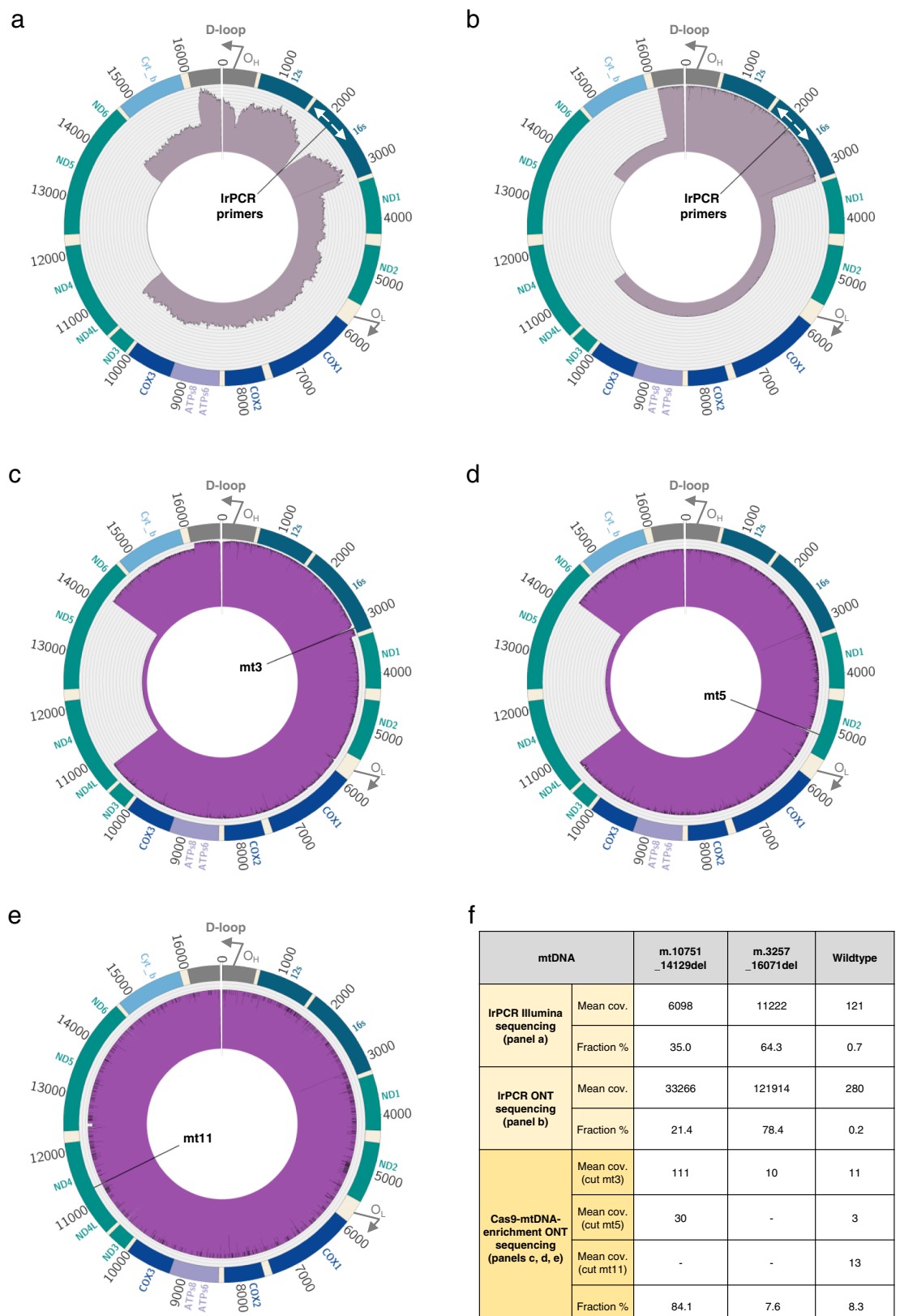

| mtDNA | | m.10751_14129del | m.3257_16071del | Wildtype |
|---|---|---|---|---|
| **lrPCR Illumina sequencing (panel a)** | Mean cov. | 6098 | 11222 | 121 |
| | Fraction % | 35.0 | 64.3 | 0.7 |
| **lrPCR ONT sequencing (panel b)** | Mean cov. | 33266 | 121914 | 280 |
| | Fraction % | 21.4 | 78.4 | 0.2 |
| **Cas9-mtDNA-enrichment ONT sequencing (panels c, d, e)** | Mean cov. (cut mt3) | 111 | 10 | 11 |
| | Mean cov. (cut mt5) | 30 | - | 3 |
| | Mean cov. (cut mt11) | - | - | 13 |
| | Fraction % | 84.1 | 7.6 | 8.3 |

datasets. These assessments are presented in the Results section. The clinical data samples allowed assessment of the heteroplasmy detection by comparing with the known clinical results for the samples. No data were excluded from the analyses. No formal assessment of sample size was performed. The investigators were not blinded to the identity of the samples.

## Human cell lines

Cell lines HEK-293, A549, Capan-2, SH-SY5Y were originally obtained from American Type Culture Collection (ATCC); ATCC N°: CRL-1573, CCL-185, CRL-2266, HTB-80, respectively. Cells were cultured according to recommended protocols and were maintained at 37 °C in 5% $CO_2$. DNA was extracted from pellets containing 2-3 million cells

**Fig. 3 | Multiple mtDNA deletions in a clinical sample. a, b** Circos plot of the lrPCR products of sample AW6506 showing three full-length lrPCR amplicons – two deletions and wild type, sequenced by Illumina short-read (**a**) and ONT long-read (**b**) instruments. White arrows at positions m.2120 and m.2119 represent forward and reverse primers, respectively. **c**–**e** Circos plots of sample AW6506 targeted with the Cas9-mtDNA-enrichment sequenced on a GridIon flow cell showing three populations of mtDNA. **c** Targeted with the gRNA mt3 (m.3127), all three populations can be observed – two deletions and the wild type; **d** gRNA mt5 (m.5142)

results in two populations – the small deletion and the wild type; **e** gRNA mt11 (m.11239) results in capturing the wild-type population only. In the circos plots, the reference circle colours denote genes encoding protein subunits of complex I (green), III (sky blue), IV (royal blue), V (light steel blue), ribosomal RNAs (ocean blue), transfer RNAs (ivory), and non-coding region D-loop (grey). **f** Summary of mean coverage of selected full-length reads and SV proportions in lrPCR amplicons, sequenced on Illumina and GridIon, and Cas9-mtDNA-enriched native molecules, sequenced on GridIon. Source data are provided as a Source Data file.

using the Nanobind CBB Big DNA kit (Circulomics) and stored at 4 °C until use.

### Clinical samples

A set of 15 clinical samples was previously collected. The DNA extraction was performed at University Hospital Vall d'Hebron (Supplementary Table 2) from blood samples using the chemagic DNA Blood400 Kit LH (Perkin Elmer). The DNA from oral mucosa swabs, urine and tissue samples was extracted using the Gentra Puregene Blood Kit (Qiagen). All subjects gave informed consent approved by the local bioethics committee and in accordance with the Declaration of Helsinki. For the purpose of clinical diagnosis the whole mitochondrial genome was amplified in a single amplicon by lrPCR using the Takara LA PCR kit (Takara) as previously described[51] using 10 ng of gDNA per reaction and mt16426F (5'- CCG CAC AAG AGT GCT ACT CTC CTC – 3') and mt16425R (5'- GAT ATT GAT TTC ACG GAG GAT GGT G – 3') as the forward and reverse primers in a 50 µl PCR reaction volume. The lrPCR conditions were: 2 min at 95 °C followed by 30 cycles of 20 s of denaturation at 95 °C and 18 min of annealing and extension at 68 °C. The final extension was 68 °C for 20 min. Qubit 2.0 Fluorometer (Thermo Fisher Scientific) was used to quantify the amplicons and each sample normalised to 0.2 ng/µl. Amplicons were fragmented using NEBNext dsDNA Fragmentase (NEB), and a library was prepared using the NEBNext Ultra II DNA library prep kit for Illumina (NEB) following the manufacturer's instructions. The pooled, indexed libraries were loaded into the MiSeq Reagent kit V2, 300 cycles (Illumina) and sequenced on the MiSeq platform (Illumina) in paired-end mode with a read length 2 × 151 bp. In general, 40 libraries are multiplexed to obtain 5000× mean coverage. In the mtDNA pipeline, quality filtering and trimming of overrepresented sequences was done using Trimmomatic[52] to eliminate the sequences corresponding to the adaptors used. After trimming, the reads were mapped against the GRCh38 reference genome using BWA-MEM[53]. For variant calling and annotation we used Pisces[54] with the following parameters: Depth ≥ 10, variant q-score between 20 and 100, variant frequency ≥0.01 and base quality ≥20.

### DNA quality control

DNA was quantified using a Qubit fluorometer with the dsDNA Broad Range Assay kit (Thermo Fisher Scientific) following the manufacturer's instructions. DNA purity was evaluated using Nanodrop 2000 (Thermo Fisher Scientific) UV/Vis measurements. To determine the cell line gDNA integrity pulse-field gel electrophoresis, using the Pippin Pulse (Sage Science) was performed. For this analysis a Sea-Kem® GOLD Agarose 1% (Lonza) gel was prepared in 0.5× TBE buffer (Thermo Fisher Scientific). Approximately 150 ng of DNA sample was loaded together with CHEF DNA Size Standard Ladder (BIO-RAD) and Quick-Load 1 kb Extend DNA Ladder (NEB) to aid size determination. Fragments were separated using a pre-set 5–80 kb protocol. After the run, the gel was visualised using a NuGenius imaging system (Syngene). DNA integrity of the clinical samples was assessed with the Femto Pulse System using the Genomic DNA 165 kb Kit (Agilent Technologies) following the manufacturer's protocol. The gDNA samples were stored at 4 °C.

### Long-range PCR

Full-length mtDNA amplification was carried out for short-read and long-read sequencing, and as a part of quality control using Platinum SuperFi II DNA Polymerase (Invitrogen). A forward primer Mt2120F (5'- GGA CAC TAG GAA AAA ACC TTG TAG AGA GAG –3') and a reverse primer Mt2119R (5'- AAA GAG CTG TTC CTC TTT GGA CTA ACA –3') were used. All lrPCRs (50 µl) were prepared as follows: 1× SuperFi II buffer, 0.4 mM each of dNTPs, 0.2 µM of each forward and reverse primers, 1× of SuperFi II DNA Polymerase, 0.5 mM MgCl₂, and 10–100 ng of individual DNA sample. Cycling conditions were: 94 °C for 1 min; 30 cycles of 98 °C for 10 s, 68 °C for 16 min; 72 °C for 10 min; hold at 4 °C. To visualise amplicons, Ultra Pure Agarose 1% (Invitrogen) gel was prepared in 1× TAE buffer (BIO-RAD). DNA samples were loaded together with Quick-Load 1 kb Extend DNA Ladder (NEB) to aid size determination. Fragments were separated running a gel at 60 V for 2.5–3 h. Amplicons were purified with AMPure XP beads (Agencourt, Beckman Coulter) and eluted in 30 µl nuclease-free water. Concentration was measured using a Qubit fluorometer with the dsDNA Broad Range Assay kit (Thermo Fisher Scientific) according to the manufacturer's instructions.

### Preparation of Cas9 ribonucleoprotein complexes

To perform a Cas9-mtDNA-enrichment of full-length mtDNA sequences on single or multiplexed up to 4 samples per flow cell, a modified Cas9 protocol was established using custom crRNAs (Supplementary Table 1). A respective number of crRNA:tracrRNA duplex pairs were prepared. Here, 10 µM of every crRNA in TE pH 7.5, (IDT) was annealed with 10 µM of tracrRNA (IDT) separately in Nuclease-free Duplex Buffer (IDT) by denaturation at 95 °C for 5 min and cooling down to RT for 10 min. Then, 10 µM of each crRNA:tracrRNA duplex was assembled together with 0.5 µM of Alt-R® S.p. HiFi Cas9 Nuclease V3 (IDT), 1× CutSmart Buffer (NEB) in nuclease-free water to form ribonucleoprotein complexes (RNPs). This mix was incubated at RT for 30 min. After incubation, formed RNPs were held at 4 °C until further use.

### Circular mtDNA pre-enrichment

To digest linear DNA 1 µg of DNA sample was incubated in 50 µl with 1× NEBuffer 4 (NEB), 1 mM ATP (NEB) and 10 units of Exonuclease V (NEB) in nuclease-free water. The reaction was incubated at 37 °C for 30 min and heat inactivated at 70 °C for 30 min. Each treated DNA sample was cleaned-up with AMPure XP beads (Agencourt, Beckman Coulter) and eluted in 24 µl nuclease-free water.

### gDNA dephosphorylation

The gDNA dephosphorylation was carried out in a total volume of 30 µl containing 1 µg DNA, 1× CutSmart Buffer (NEB) and 15 units of Quick CIP (NEB) by incubating at 37 °C for 20 min followed by Quick CIP inactivation at 80 °C for 3 min. For the following Cas9-mtDNA-enrichment procedure each DNA sample was split into aliquots.

### Cas9-mtDNA-enrichment, library preparation, multiplexing and long-read sequencing

Each dephosphorylated DNA sample aliquot (15 µl) was mixed with 5 µl of appropriate RNP complex, and incubated for 20 min at 37 °C. The mtDNA digestion was followed by incubation with 0.12 units of

Thermolabile Proteinase K treatment (NEB) for 15 min at 37 °C and 10 min at 55 °C for inactivation. The dA-tailing was done by adding 0.25 mM dATP (NEB), and 0.5 µl Taq polymerase (NEB) for 10 min incubation at 72 °C. Following dA tailing, all processed DNA aliquots were pooled together. The DNA sample pool was purified with AMPure XP beads (Agencourt, Beckman Coulter) and eluted in 42 µl nuclease-free water and processed with Ligation Sequencing Kit, SQK-LSK112 (ONT). The ligation mix was prepared in 38 µl containing 20 µl of Ligation Buffer (ONT), 5 µl of Adapter Mix H (ONT), 10 µl of T4 DNA ligase (NEB) and 3 µl of nuclease-free water. The ligation mix with the library was incubated for 1 h at RT. Ligation was terminated with TE buffer, pH 8, the sample incubated with AMPure XP beads (Agencourt, Beckman Coulter) and washed twice with short fragment buffer (ONT). The sequencing flow cell was prepared according to ONT recommendations and Loading Beads II (ONT) were added to the library immediately prior to loading of the sample.

The library was sequenced on GridIon Mk1 using the R9.4.1, R10.3 or R10.4 flow cells (ONT). Prior to every experiment, quality control was performed for each GridIon flow cell using the MinKNOW software (v.21.05.20-21.10.8). Sequencing experiments were running for 72 h with off-line basecalling.

## lrPCR amplicons long-read nanopore sequencing

Library preparation was started with DNA repair and end-prep of 50 fmols of lrPCR mtDNA amplification product using NEBNext FFPE DNA Repair Buffer (NEB), NEBNext FFPE DNA Repair Mix (NEB), Ultra II End-prep reaction buffer (NEB) and Ultra II End-prep enzyme mix (NEB) incubated at 20 °C for 30 min and at 65 °C for 30 min according to ONT protocol. Each sample was purified with AMPure XP beads (Agencourt, Beckman Coulter) and eluted in 60 µl nuclease-free water. For adapter ligation, elution was mixed with 25 µl Ligation Buffer (ONT), 10 µl NEBNext Quick T4 DNA Ligase (NEB) and 5 µl Adapter Mix H (ONT). The reaction was incubated for 1 h at RT. Ligation was terminated with a clean-up using 0.4× volume of AMPure XP beads (Agencourt, Beckman Coulter). The library was quantified using a Qubit fluorometer with the dsDNA Broad Range Assay kit (Thermo Fisher Scientific). Approximately 10 fmol of library was loaded on the R10.4 flow cell and sequenced on GridIon Mk1 following manufacturer's recommendations. Sequencing experiment was set to run for 100 h with off-line basecalling.

## lrPCR amplicons short-read sequencing

Short-insert paired-end libraries of the lrPCR mtDNA product of the cell lines were prepared with KAPA HyperPrep kit (Roche) with some modifications. In short, 1 µg of lrPCR DNA was sheared on a Covaris™ LE220-Plus (Covaris). The fragmented DNA was end-repaired, adenylated and Illumina platform compatible adaptors with unique dual indexes and unique molecular identifiers (IDT) were ligated. The libraries were quality controlled on an Agilent 2100 Bioanalyzer with the DNA 7500 assay for size and the concentration was estimated using quantitative PCR with the KAPA Library Quantification Kit Illumina Platforms (Roche).

The library preparation from the lrPCR product originating from the clinical samples was carried out with Illumina DNA PCR-Free Library Prep, Tagmentation Kit (Illumina) following the Illumina reference guide instructions and recommendations. For each sample, 100 ng of lrPCR mtDNA product was tagmented, purified and ligated to IDT-ILMN UD indexes (Illumina). Samples were pooled by volume using an index correction factor (from Illumina technical note "Balancing sample coverage for whole-genome sequencing and its associated index correction values"). Library pools were quantified with Qubit ssDNA (single-stranded) assay (Invitrogen) and molarity values were calculated considering 450 bp as the average library size.

Paired-end DNA sequencing (2 × 151 bp) of the libraries was performed on an Illumina NovaSeq 6000 sequencer using the NovaSeq S4 Reagent Kit, v1.5; 300 cycles following the manufacturer's protocol. Image analysis, basecalling and quality scoring of the run were processed using the manufacturer's software Real Time Analysis (RTA v3.4.4) and followed by generation of FASTQ sequence files.

## Pre-processing raw data

At the time of the analysis, the pre-processing protocol of the raw Oxford Nanopore reads produced with the Q20+ ligation sequencing kit was still under development (https://nanoporetech.com/about-us/news/new-nanopore-sequencing-chemistry-developers-hands-set-deliver-q20-99-raw-read), so we used a custom snakemake pipeline[55] (https://github.com/marcDabad/q20plus_rebasecall), which involves four different tools. First, the basecalling of the simplex data was performed using Guppy v5.0.16 (https://nanoporetech.com/community). Then, the Duplex Sequencing Tools v0.2.3 (https://github.com/nanoporetech/duplex-tools/) was used to identify and filter the paired duplex reads. Following, the putative duplex reads filtered were high quality basecalled using Guppy Duplex Basecalling beta v0.0.0 (https://nanoporetech.com/community). The last step consisted of merging and removing the redundancy from the two basecalling data results. For this task, we applied an in-house developed tool called fastq_merge (v0.3.0). It was noted that (a) the % of duplex reads was very low (-1%) and (b) the base qualities of the reads were wrong with all bases having the same, very high, base quality (61). To avoid any potential biases introduced by such reads it was therefore decided to eliminate the duplex reads from the subsequent analyses. Two models to perform the basecalling were applied: res_dna_r9.4.1_e8.1_sup_v033.cfg (from the Rerio Database) for the FLO-MIN106D (R9.4.1) flow cell generated data, and dna_r10.4_e8.1_sup.cfg (from Guppy) for the FLO-MIN112 (R10.4) data. Quality control checks were performed with MinIONQC.R[56] which uses the final sequencing_summary.txt file.

## Alignment and variant calling

Alignment of nanopore reads for demultiplexing was performed using minimap2 v2.24-r1122 to version GRCh38 of the human genome producing a PAF alignment file. The FASTQ files were demultiplexed into individual files per cut-site using ont_demult v0.3.3 (developed at the CNAG-CRG), which takes as input the original FASTQ files, the PAF alignment file from minimap2 and a configuration file that gives the location of the cut-sites and the correspondence between sample identifiers and cut-sites. The demultiplexed files were then re-aligned using minimap2, this time to a custom reference derived from GRCh38 that was specific for each cut-site.

Let R represent the original mitochondrial sequence and $S_i$ the custom sequence for cut-site i, and let l be the length of R in base pairs. The length of $S_i$ is $2(l + d)$ for all i, where d is a short distance (5 bp was chosen for the analysis presented here) to allow for some imprecision in the mapping. The general structure of each $S_i$ is the same: the first d bases are from the last d bases of the mitochondrial reference. These are followed by 2 complete copies of the mitochondrial reference, followed finally by the first d bases of the reference. Assuming that a cut-site does not occur in the first or last d bases of R, each cut-site is represented twice in S. For example, a cut-site mapping to position x in R will map to $x + d$ and $x + l + d$. If all reads were close to full length then each read would map uniquely to S, however shorter reads could map twice. This is why a separate reference is made for reads allocated to each cut-site. For each $S_i$, all bases except for the sequence from $x − d$ to $x + l + d$ inclusive are masked (set to N), thus ensuring that any read longer than d that has been allocated to cut-site i should only map once to $S_i$ (Supplementary Fig. 2). Since all $S_i$ have the same structure, reads mapped to different references will have mapping positions that correspond (modulo l). The BAM files corresponding to each sample (from different cut-sites) were merged to produce the sample BAM files.

Variant calling was performed using baldur v1.1.8 (developed in-house) that uses a maximum likelihood framework to determine the

most likely alleles at each site as well as their respective VAF. Unlike variant calling from genomic data where for autosomes it is expected that there are 1 or 2 distinct alleles per site with frequencies of heterozygote sites at 50%, with mitochondrial data there are no a priori expectations for either the number of alleles or the VAF at each site. In this respect the analysis of mitochondrial variants resembles more the analysis of somatic variants in cancer samples or pooled population samples than that of autosomal genomic variants. Potential alleles at single or multibase sites are identified as having >1 observation on each strand. An error model is used for each observation, so that for an observed allele $a$ with VAF $f_i$ and expected error probability $e_j$, the contribution to the likelihood from this observation $p(f_i, e_j) = f_i(1 - e_j)(1 - f_i)e_j$. For single bases, $e_j$ is taken from the base quality value $q_j$ where $e_j = e^{-0.1 q_j ln(10)}$. For indels the quality of the preceding base is used, and for multibase alleles (including insertions) the lowest quality value of the component bases is used. Let $c_{i,j}$ be the observed counts of allele $i$ with quality level $j$, and $e_j$ the error probability of observations at quality level $j$. The overall log likelihood is: $L(f) = \sum_{i,j} c_{i,j} ln(p(f_i, e_j))$.

Fisher's scoring method[57] is used to obtain the maximum likelihood estimates for $f$. Since all frequencies have to sum to 1, if we have $m$ alleles we have $m - 1$ parameters to maximise. We can therefore define $f_0 = 1 - \sum_{i>0} f_i$. For the scoring method we need the score vector $\mathbf{S}$ or vector of the partial differentials $\frac{\partial L(f)}{\partial f_i}$ and Fisher's information matrix $\mathbf{I}$. The score vector can be obtained for $f_i$ where $i > 0$ from $\frac{\partial L(f)}{\partial f_i} = \sum_j (1 - 2e_j) \left( \frac{c_{i,j}}{p(f_i,e_j)} - \frac{c_{0,j}}{p(f_0,e_j)} \right)$.

Let $c_i$ be the total number of observations at quality level $j$. The diagonal elements of $\mathbf{I}$ are then $I_{i,i}(f) = \sum_j (1 - 2e_j)^2 c_j \left( \frac{1}{p(f_i,e_j)} + \frac{1}{p(f_0,e_j)} \right)$ and the off-diagonal elements are all identical and are given by $I_{i,j}(f) = \sum_j (1 - 2e_j)^2 \frac{c_j}{p(f_0,e_j)}$.

Given an initial starting vector of VAF obtained from the raw allele proportions, a new update can be obtained from $f' = f + aI^{-1}S$ where $a$ is between 0 and 1 chosen to ensure (a) that the likelihood does not decrease and (b) that all frequencies are ≥0. If the update moves a VAF to 0 the score vector is re-evaluated. If the corresponding partial derivative is negative (indicating that the maximum likelihood estimate for that VAF is negative), the VAF is fixed at zero and the corresponding allele is removed from the maximisation procedure. If no value of $a > 0$ exists that leads to an increase in the likelihood the maximisation process is stopped. At this point the variance of the VAF estimates of the remaining alleles are estimated from the inverse of $\mathbf{I}$. Likelihood ratio tests for each remaining allele in the model are carried out by fixing the VAF of each allele in turn to 0 and maximising the VAF of the other alleles.

Short-read data were processed following the Broad Institute's best practices for variant calling of mitochondrial variants. Briefly, starting from a whole genome alignment, those reads mapping to chrM were extracted and aligned with BWA-MEM to a reference containing only chrM sequence or chrM shifted 8 kb. Variant calling was performed on both alignments with mutect (GATK v.4.1.7.0) and the variants on the shifted alignment were lifted over to the regular chrM sequence.

### Annotation of mtDNA variants

Variants called were converted to VCF format with bcftools[58]. Heteroplasmy fraction of each variant was computed and appended to the VCF file with a custom script. Briefly, heteroplasmy fraction in a given position was computed by dividing the sum of reads supporting the alternative allele by the total number of reads spanning that position. Sample major and minor haplogroups were determined with Haplocheck[46].

Variants were annotated with functional annotations, population frequencies and pathogenicity predictors with SnpEff[59] and SnpSift[60].

gnomAD[44], MITOMAP[43] and MitoTIP[44,45] were used as sources of annotation. Relevant information fields of each variant were extracted and converted to a tabular format with bcftools.

Possible disease-associated variants were prioritised based on heteroplasmic fraction, population frequencies and "Confirmed" or "Reported" status in MITOMAP. Variant filtering was performed with bcftools.

### Circos plots

To prepare the plots, reads were mapped using the software Minimap2 (2.24). Then reads mapping to chrM were extracted and the coverage was computed for each locus using samtools depth (v. 1.14). The output was parsed to create appropriately formatted files for input to Circos[61] (v. 0.69-9).

### Reporting summary

Further information on research design is available in the Nature Research Reporting Summary linked to this article.

## Data availability

The clinical and the cell line sequencing data generated in this study have been deposited in the European Genome-phenome Archive (EGA) under the accession codes EGAS00001006203 and EGAS00001006280, respectively.

The sequencing data used in our methodology for targeting, sequencing and multiplexing single molecules for mitochondrial variant research are under restricted access in order to protect the sample donors. Researchers interested to access the data must apply directly to the Data Access Committees through the EGA or by direct e-mail.

Request for data access directly to the Data Access Committee should be done through:

https://ega-archive.org/dacs/EGAC00001002647
https://ega-archive.org/dacs/EGAC00001002698

Source data are provided with this paper. The sequencing depth data used in this study for Fig. 3 and Supplementary Fig. 1, and the correlation between the mtDNA heteroplasmy frequencies in cell lines measured by lrPCR with Illumina sequencing and Cas9-mtDNA-enrichment with ONT sequencing in Supplementary Fig. 3 are provided in the Source Data file. Source data are provided with this paper.

## Code availability

The three utilities[62–64] developed locally and used in this manuscript are freely available from GitHub repository:

baldur, v1.1.8: https://github.com/heathsc/baldur.git.
ont_demult, v0.3.3: https://github.com/heathsc/ont_demult.git.
fastq_merge, v0.3.0: https://github.com/heathsc/fastq_merge.git

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

## Acknowledgements

This research has received funding from the European Union's Horizon 2020 research and innovation programme under grant agreement No 824110 – EASI-Genomics (I.G.G.) and the ERC Synergy project BCLL@las under grant agreement No 810287 (I.G.G.). Institutional support was from the Spanish Instituto de Salud Carlos III, Fondo de Investigaciones Sanitarias and cofunded with ERDF funds (PI19/01772). We acknowledge the institutional support of the Spanish Ministry of Science and Innovation through the Instituto de Salud Carlos III and the 2014–2020 Smart Growth Operating Program, to the EMBL partnership and institutional co-financing with the European Regional Development Fund (MINECO/FEDER, BIO2015-71792-P). We also acknowledge the support of the Centro de Excelencia Severo Ochoa, and the Generalitat de Catalunya through the Departament de Salut, Departament d'Empresa i Coneixement and the CERCA Programme to the institute.

## Author contributions

M.G. and I.G.G. conceived the project. I.K., P.B., M.G. and I.G.G. developed the method. I.K. performed experiments. I.B.H. provided cell line samples. A.A. and E.G.A. contributed to patient recruitment and provided clinical samples; J.L. and E.G.A. processed and analysed the clinical samples at the diagnostic laboratory. S.C.H. developed the statistical analysis and the informatics pipeline. S.C.H., D.C., C.F.L., M.D., R.T.H., I.P., M.J.I. processed and analysed the data. I.K., S.C.H., M.G. and I.G.G. wrote the manuscript. All authors revised and approved the manuscript for publication.

## Competing interests

The authors declare no competing interests.
