## [Peer Review File · Nature Communications]

REVIEWER COMMENTS

Reviewer #1 (Remarks to the Author):

This article presents an innovative long-read sequencing approach to capture and sequence at high depth full length mitochondrial genomes as native single-molecules, utilizing the RNA-guided DNA endonuclease Cas9. Furthermore, the authors present an informatics based approach to demultiplex, align and annotate this data.

The article is well written, and figures and tables are clearly presented. Furthermore, the work presented strongly supports the conclusions drawn by the authors. The limitations to current amplification and short-read based sequencing approaches that can be utilised to analyse the mitochondrial human genome are clearly explained. The methods are detailed and offer readers enough detail to generate reproducible data. The application of this current technology to mtDNA disease is also well contextualised. I am excited by the potential clinical applications of this approach and congratulate the authors on their development of this method and custom analysis pipeline.

Minor points

The authors note that enrichment methods to favor adaptor ligation at the cut-sites of RNA-guided Cas9 have in the past been combined with both ONT and PacBio long-read-sequencing platforms. They then go on to highlight that the relatively high error rate of ONT sequences makes accurate detection of SNVs and short indels challenging, despite specialist tools being developed. I am intrigued to know why they favor the combination of ONT over PacBio long-read sequencing approaches, given the PacBio platform is established to have lower error rates and at therefore at face value seem a better choice for accurate detection of SNVs and indels. It would perhaps be nice if they could comment on this in their discussion so readers could better understand the logic of their choice of ONT over PacBio. My guess is the ONT platform overall is much more affordable, which is a strong reason in itself, but nonetheless it would be nice for the authors to explain.

Given the protocol sequencing native DNA molecules (independent of any amplification step) I assume that high quantities + quality of DNA are required as input and this may limit the clinical applications. Are the authors able to comment on this point? Furthermore, do the authors envisage that the cost of this approach will limit its diagnostic utility?

It would also be helpful if the authors could contextualise if the methods/tools developed have potentially any broader applicability beyond mt Human disease in their discussion.

Reviewer #2 (Remarks to the Author):

Keraite and colleagues describe a method to perform long read sequencing of the mitochondrial genome (mtDNA), based on Oxford Nanopore Technology. They describe an experimental protocol to (i) optimize selection of mtDNA from samples (Exonuclease V to selectively degrade nuclear DNA), (ii) linearize and barcode mtDNA (using Cas9 with several dual-guide RNAs targeting each sample), (iv) further optimization (Proteinase K to remove the Cas9 protein), (iii) multiplex sequence with long reads (ONT 12 Chemistry), and (v) analyze the data (custom pipeline to demultiplex samples, map, and call SNV and deletions). They assess the accuracy of their pipeline compared to the standard Illumina sequencing with GATK analysis pipeline, and present results on patient samples with pathogenic SNVs and one case with a complex deletion. The example with the complex deletion clearly shows the advantage of their technology over existing technologies for deletion analysis.

This work is novel, innovative, and important. Using Cas9 to simultaneously linearize and barcode samples is clever. If the method is robust and accurate it would represent a key advance for the field. However there are several major critiques that need to be addressed before it is clear how robust and accurate the method actually is. In addition, there are several minor critiques that should be addressed to clarify the description of the method and fix inaccuracies.

Major critiques:

- The presented data show that even after optimization, relatively few ONT long-reads actually sequence the entire mtDNA molecule and map correctly to both the Cas9 break start and stop (eg Fig 3f). The authors interpret this as a problem of sample degradation with “data not shown”. This is a major issue and deserves to be explored more fully since it has enormous implications for how/when this method will work robustly. If it is indeed a problem of degradation, I would expect fresh DNA (from cell lines or blood) would not show this problem. Is that the case? Even under optimal sample preparation conditions (Table S6) only ~35% of reads mapping to the mtDNA are full-length with “Both” ends mapping. Is this a problem with ONT more generally, or specific to mtDNA, or to sample degradation? These issues need to be addressed and may require more sequencing of fresh samples for validation.
- How accurate is the method compared to the current standard (either IrPCR Illumina or short read Illumina)? The authors do multiple experiments but do not summarize the results in terms of sensitivity and precisions (eg what % of homoplasmic variants (VAF -.95-1.00) are shared between Illumina & ONT vs technology-specific? At different heteroplasmy levels (eg VAF 0.05-0.95, or VAF 0.01-0.05) what % of variants are shared between technologies? Accuracy is a huge issue that needs to be properly addressed, and may require additional ONT sequencing – eg on HapMap or cancer cell lines for which Illumina WGS data are already available so that a head-to-head comparison can be made.

- How much are NUMTs a problem for this pipeline? The authors discuss that NUMTs cause problems for short read methods, but do not address how much a problem they are for their method. There are two types of NUMTs that need to be considered: those that are in the reference genome (eg on chr1 and chr17), and those that are not in the reference genome and are polymorphic across humans. Previous studies (such as the gnomAD analysis) show that for short Illumina reads, polymorphic NUMT misalignment causes enormous problems for variants with heteroplasmy < 5% in blood. Ideally long-reads from ONT should solve this problem, especially if the “Both” strategy is applied – but the authors do not discuss issue this nor provide evidence that their method avoid NUMTs. They should also discuss the failure scenario of the same type that caused the extremely controversial paper “Biparental inheritance of mtDNA in humans” (PMID: 30478036), which is most likely caused by fairly rare tandemly repeated NUMTs termed “mega-NUMTs” – which would presumably also be problematic for this ONT pipeline.

Minor critiques:

- GeneBank should be GenBank (both in text and supplemental tables)
- Table S8 is inaccurate with gnomAD allele frequencies – whereby homoplasmic and heteroplasmic allele counts (and allele frequencies) are swapped. EG pathogenic m.3243 is detected in 6 individuals at heteroplasmy and 0 at homoplasmy in gnomAD v3, however Table S8 lists 0 heteroplasmies and 6 homoplasmies – compromising data interpretation. Please check accuracy of all column headers.
- Method for demultiplexing and then aligning to custom reference sequence is confusing. The method describes a two step process: (A) align to chrM, (b) demultiplex, (c) realign to a tandem chrM:chrM concatemer sequence with per-gRNA masking. Why are two different mappings required? It was unclear why a single mapping to a tandem chrM:chrM wouldn’t allow all long reads could properly align across the artificial origin, and why masking was needed (since the different gRNA could be remapped in the pileup stage using their strategy of $x \bmod 16569$).
- The authors do not discuss if their selected gRNA sequences are polymorphic (eg common polymorphisms in GenBank or gnomAD)

Exposition in the manuscript

- The advances of this pipeline can be more clearly described to explain the actual experimental and computational pipeline. For example, the optimized protocol should be clearly explained (eg Exonuclease V only explained in supplement not in manuscript, and it’s not clear where in the experimental pipeline this is required; similar for Proteinase K). The advances to optimize the protocol are important innovations, that are currently be lost/buried in the text.
- Figure 1: it’s unclear that multiple gRNAs are targeting the same sample prior to multiplexing.
- Figure 2: Mitogenome is non-standard term. Perhaps mtDNA or chrM?

- Figure 2 and text: "Frequencies of alleles" is unclear. Often "allele frequency" indicates population allele frequency (eg MAF =minor allele frequency across individuals). I think you're talking about Variant Allele Fraction (sometimes termed VAF), which is the fraction $\text{ALT_reads}/(\text{ALT_reads}+\text{REF_reads})$.
- Table 1: it is unclear what that clinical sequencing lab's heteroplasmy was. Is this the "IrrPCR Illumina" columns? Or did you do your own IrrPCR and sequencing?

Reviewer #3 (Remarks to the Author):

In their study, Keraite et al. present a novel method for enriching for mtDNA molecules from a complex DNA source using the DNA endonuclease Cas9 and direct sequencing of long DNA molecules on the Oxford Nanopore platform. While the enrichment method using Cas9 itself is not novel, its application to the mtDNA genome is and the authors can convincingly show that it circumvents some common issues when analysing mtDNA heteroplasmies with Illumina short-read data, such as the PCR amplification bias and the issues of phasing mutations that are further apart than a few hundred base pairs. The manuscript is very well written and one can easily follow their line of argument. However, I have a few questions and comments:

- The authors used for their experiments the new ONT R10.4 flow cells that help to reduce the sequencing error but reported in their discussion (l. 338ff) that they obtained very similar results when using the older, still very common ONT R9.4 flow cells that have a higher sequencing error rate (Supp Table 5 & 6). However, the results for the ONT R9.4 flow cell differ very much between the two tables. In Supp Table 5, the fraction of chrM reads was just 23% and as low as the values that were obtained for the ONT R10.4 flow cells without enrichment. In Supp Table 6, the fraction of chrM reads is suddenly 60%. The same goes for the average divergence, which drops from 1.8% in Supp Table 5 to 1.1% in Supp Table 6. What's the difference between these results presented in these tables? Furthermore, neither of these tables is referenced in the main text, just in the Supplementary Material. The authors should add a reference to these tables in the discussion.

- Their new program baldur is able to phase SNVs over longer regions than it is typically possible using only Illumina short-read data (l. 232ff.). While the authors show that this works very well for a cell line sample and high frequency heteroplasmies (Supp Figure 4), they do not mention how they consider divergent sequences caused by sequencing errors or differences in sequencing length as they are common for clinical samples. It would be interesting to know in which way these type of uncertainties of the actual mtDNA haplotype are considered during phasing.

- When studying a clinical sample with a large deletion in the mtDNA (AW6506) that could not be resolved using short-read sequencing, the authors used three guide RNA pairs to prepare the DNA for

sequencing. Depending on the location of the cut sites, they were able to detect of all different haplotypes (wildtype plus two deletions; Figure 3 c-e). This highlights the effect of the positioning of the cut sites for the success of recovering all possible haplotypes in a sample. While the authors stated that there was not correlation between the location of the cut sites and obtained coverage along the mtDNA genome (l. 321ff.), I couldn't find any data that could actually show this. For the cell line results, each cell line had their specific cut site and with the exception of the sample AW6506, there were no replicates of the same samples with different gRNAs. It would be interesting to see if the authors could actual show this experimentally by applying all four pairs gRNAs to the same cell line and comparing the resulting coverage and variance of the coverage across the mtDNA genome.

- As a minor comment, in Supp. Table 8, the columns "Heteroplasmic_fraction" and "Heteroplasmy_%" are identical and therefore redundant; furthermore, I don't think the label "Heteroplasmic_fraction" is the correct term of what is measured because a Heteroplasmic fraction

We thank all Reviewers for their valuable comments and thoughtful insights that have enabled us to improve our manuscript. Minor changes to the manuscript have been made using Track Changes, while major re-writes are highlighted in yellow.

Reviewer #1:

R1: Why authors favor the combination of ONT over PacBio long-read sequencing approaches

The cost of reagents, capital cost of instrumentation and broader availability of ONT was the main motivation of our choice; however, the method could be applied on PacBio as well, if that was the available platform.

R1: Given the protocol sequencing native DNA molecules (independent of any amplification step) I assume that high quantities + quality of DNA are required as input and this may limit the clinical applications. Are the authors able to comment on this point?

It is not necessary to use special ultra HMW DNA extraction kits usually used for long-read sequencing. The quality obtained from widely used gDNA extraction kits, such as from Qiagen, is sufficient. Further, although it is preferable to use DNA with good integrity to receive a high proportion of full-length reads, with minor changes, our pipeline is also suitable for more degraded DNA samples and still provides accurate results. Non full-length reads ('Start') can be used to boost coverage. The clinical samples in our experiments were processed without the Exo V step as DNA quality was not optimal. For a standard reaction we used 500 ng of gDNA for each Cas9-mtDNA-enrichment aliquot. For the multiple deletion sample (AW6506) we had to go to 300 ng for each Cas9-mtDNA-enrichment aliquot due to limited material availability (see Fig. 3). Importantly, we could demonstrate it was possible to use non-invasively collected DNA sample types, such as from urine and buccal swabs. Non-invasive specimens are favored in the diagnostic environment and DNA extraction protocols for such specimens are well established in clinical laboratories.

R1: Do the authors envisage that the cost of this approach will limit it's diagnostic utility?

This procedure is cost efficient, and multiplexing more samples into one flow cell reduces the cost further. The pipeline is straightforward from sample preparation to analysis, which can be done on a laptop computer. Clinical laboratories use a variety of different strategies to characterize the mtDNA status of a

sample, while our approach comprehensively covers all aspects we tested and is simple.

R1: It would also be helpful if the authors could contextualise if the methods/tools developed have potentially any broader applicability beyond mt Human disease in their discussion.

We appreciate that the Reviewer sees the potential of our method beyond the analyses of mtDNA in the context of human mitochondrial disease. We also expect this method will open a new avenue for mtDNA research in other species. We have modified and expanded the Discussion accordingly. As we already mention in the manuscript, we see potential for the investigation of somatic mutations in cancer in ChrM by our method. As also pointed out it is applicable for human population studies.

Reviewer #2:

R2: The presented data show that even after optimization, relatively few ONT long-reads actually sequence the entire mtDNA molecule and map correctly to both the Cas9 break start and stop (eg Fig 3f). The authors interpret this as a problem of sample degradation with “data not shown”. This is a major issue and deserves to be explored more fully since it has enormous implications for how/when this method will work robustly. If it is indeed a problem of degradation, I would expect fresh DNA (from cell lines or blood) would not show this problem. Is that the case? Even under optimal sample preparation conditions (Table S6) only ~35% of reads mapping to the mtDNA are full-length with “Both” ends mapping. Is this a problem with ONT more generally, or specific to mtDNA, or to sample degradation? These issues need to be addressed and may require more sequencing of fresh samples for validation.

We apologize for having created some confusion for the reviewer. The issue of DNA degradation and a less high fraction of full length reads (“both”) refers to DNA extracted from the oral mucosa and urine samples – which is also where we used the statement “data not shown”. We have now included the quality control results showing the gDNA integrity in the Supplementary material and substituted the statement “data not shown” with Supplementary Figure 5.

In sample AW6506, the muscle biopsy with multiple deletions, the gDNA is of decent quality (Supplementary Figure 5). However, due to the limited amount of sample available to us, we had to decrease the amount of input gDNA for each of the three Cas9-mtDNA enrichment aliquots used to 300 ng. In Figure 3 we show the circos plots with the full length reads only (“both”), as this graphically shows the proportions of the three mtDNA species in this sample. Clearly this demonstrated the superiority over methods applying amplification by IrPCR. Obviously applying a cut outside of the major species, in an underrepresented species results in a reduced number of reads of the underrepresented species (Figure 3d, with 2 species and Figure 3e with only the wildtype species). Nonetheless the overall coverage with full length reads is >100x. The results from the three aliquots give very consistent results which is a nice internal control.

It is important to point out that for the robustness of calling and reporting of single nucleotide and structural variants, baldur always uses the full length reads (“both”) and the fragmented reads (“start”). The boost of coverage increases robustness (see Table1).

We extensively tested freshly extracted DNA from four different cell lines (Supplementary Table 5 and 6). The fraction of mtDNA mapping reads of cell line DNA samples was 50-60%, of which more than half reads were full length (selected with the most stringent strategy 'Both'). Avoiding breaking molecules during ONT library preparation is not possible due to pipetting steps involved. Other Cas9 targeted enrichment ONT publications show a highest fraction of on-target reads of 5% of total reads (not necessarily full-length; Gilpatrick et al. 2020, Iyer et al. 2022). Therefore, our mtDNA enrichment is superior in terms of Cas9 ONT enrichment approaches. We have also looked at a recent presentation from Roche comparing different short read enrichment methods (<https://sequencing.roche.com/content/dam/rochesequence/US/Teaser/Images/TargetEnrichment/kapa-te-base-coverage-fig-A.png>). For the most commonly used exome enrichment methods, they report 68-80% reads on target, keeping in mind that the Roche definition “% reads on-target” includes “all sequence reads that overlap the target region by even only a single base”. A further consideration is that the target region of an exome is substantially larger than the mtDNA target.

R2: How accurate is the method compared to the current standard (either IrPCR Illumina or short read Illumina)? The authors do multiple experiments but do not summarize the results in terms of sensitivity and precisions (eg what % of homoplasmic variants (VAF -.95-1.00) are shared between Illumina & ONT vs technology-specific? At different heteroplasmy levels (eg VAF 0.05-0.95, or VAF 0.01-0.05) what % of variants are shared between technologies? Accuracy is a huge issue that needs to be properly addressed, and may require additional ONT sequencing – eg on HapMap or cancer cell lines for which Illumina WGS data are already available so that a head-to-head comparison can be made.

We apologize that this important metric was not included in the main text, this has been corrected. For all variants with $VAF \geq 0.05$ the Illumina and ONT results were 100% concordant, so if we treat the Illumina results as the truth, the ONT pipeline has a sensitivity and precision of 1 for all such variants. We get an average sensitivity of 0.80 and precision of 0.91 for low VAF variants (heteroplasmy from 0.5% - 5%), and an average sensitivity and precision of 0.88 and 0.95 for all heteroplasmic variants with $VAF \geq 0.5\%$. We described this in the Supplementary Material (Validation of the variant calling pipeline...). The result has been included in the main manuscript text.

We followed the suggestion of the reviewer and have analyzed NA12878 by our Cas9-mtDNA enrichment method. Unfortunately the GIAB dataset does not contain the mitochondrial DNA analysis. Further, we expected differences in the mtDNA variant calling due to different passaging timepoints of the LCLs and

therefore identifying false discrepancies in data analysis from current Coriell gDNA variant calling and already available datasets from NA12878 or cancer cell lines. To demonstrate both we designed a benchmarking experiment of sequencing and mtDNA variant calling of the Illumina data with two different variant calling pipelines to create a gold dataset from two different timepoints of NA12878 purchase. We sequenced NA12878 gDNA, purchased from Coriell in 2015 and in 2022, WGS at 30x coverage with Illumina (using the ISO/IEC 17025:2017 accredited WGS pipeline) and Cas9-mtDNA enrichment with ONT. The Illumina data was analysed using the mitochondrial calling GATK Best Practices Pipeline (ISO/IEC 17025:2017 accredited variant calling pipeline) and also the Illumina commercial pipeline using Dragen v3.10. The Cas9-mtDNA-enrichment data was analysed using the baldur. Two heteroplasmic variants were detected (passing filters) in both samples from the Illumina data analysis: G14918A and G16023A as well as 14 homoplasmic variants. Both heteroplasmic variants and the 14 homoplasmic variants were also detected (passing filters) from our analysis of the Nanopore data. The estimated VAF from the two heteroplasmic variants are shown in the table below. It can be seen that the VAF estimates are consistent between the two technologies, and that the variance in VAF estimates between the two samples is much greater than the variance between the two technologies.

	NA12878 (2015)			NA12878 (2022)		
	Illumina		ONT	Illumina		ONT
	Dragen v3.10	GATK Best Practices	baldur	Dragen v3.10	GATK Best Practices	baldur
G14918A	2.90%	3.00%	2.50%	4.10%	4.10%	5.40%
G16023A	54.40%	53.50%	56.00%	72.40%	72.40%	72.50%

R2: How much are NUMTs a problem for this pipeline? The authors discuss that NUMTs cause problems for short read methods, but do not address how much a problem they are for their method. There are two types of NUMTs that need to be considered: those that are in the reference genome (eg on chr1 and chr17), and those that are not in the reference genome and are polymorphic across humans. Previous studies (such as the gnomAD analysis) show that for short Illumina reads, polymorphic NUMT misalignment causes enormous problems for variants with heteroplasmy < 5% in blood. Ideally long-reads from ONT should solve this problem, especially if the “Both” strategy is applied – but the authors do not discuss issue this nor provide evidence that their method avoid NUMTs. They should also discuss the failure scenario of the same type that caused the extremely controversial paper “Biparental

inheritance of mtDNA in humans” (PMID: 30478036), which is most likely caused by fairly rare tandemly repeated NUMTs termed “mega-NUMTs” – which would presumably also be problematic for this ONT pipeline.

We were considering NUMTs when designing the method and have three aspects that help us reduce the effect of NUMTs: 1) we digest linear DNA by including the ExoV reaction; 2) guide RNAs in our experiment contain several mismatches relative to the reference NUMTs on Chr1 and Chr17. This is associated with reduced efficiency of the guides in such loci; 3) reads containing any nuclear sequence are eliminated by baldur in the alignment. If there were a concatenation of the full mtDNA sequence, cutting on the level of nuclear DNA to obtain a 16.6 kb product requires two successful cuts over one that is required for the circular mtDNA.

It is possible to design an experiment to delineate this controversial issue with a minor adaptation of our method, for example by doing the analysis with and without the ExoV treatment and allowing reads larger than 16.6 kb (mapping to a concatemer of the mtDNA reference). A distortion should be easy to spot. We could not include such an experiment because the source of such material is extremely rare.

Minor critiques:

R2: GeneBank should be GenBank (both in text and supplemental tables)

It is now corrected and highlighted across the manuscript and supplemental tables.

R2: Table S8 is inaccurate with gnomAD allele frequencies – whereby homoplasmic and heteroplasmic allele counts (and allele frequencies) are swapped. EG pathogenic m.3243 is detected in 6 individuals at heteroplasmy and 0 at homoplasmy in gnomAD v3, however Table S8 lists 0 heteroplasmies and 6 homoplasmies – compromising data interpretation. Please check accuracy of all column headers.

It is now corrected and highlighted in Table S8.

R2: Method for demultiplexing and then aligning to custom reference sequence is confusing. The method describes a two step process: (A) align to chrM, (b) demultiplex, (c) realign to a tandem chrM:chrM concatemer sequence with per-gRNA masking. Why are two different mappings required? It was unclear why a single mapping to a tandem chrM:chrM wouldn't allow all long reads could

properly align across the artificial origin, and why masking was needed (since the different gRNA could be remapped in the pileup stage using their strategy of $x \bmod 16569$).

The two-step process allows the pipeline to handle full-length and shorter reads in the same way. If a single mapping to a tandem chrM:chrM reference was performed, full-length reads would indeed map correctly across the origin without artifacts, but shorter reads could potentially map twice perfectly, thus giving a MAPQ of zero. These reads could be rescued, but this would require checking all reads with MAPQ of zero to verify which of these had precisely two mappings of the form x and $x + 16569$. This is clearly possible, but it was decided to follow the simpler option of remapping (which is very fast as we are only mapping to a 32kb reference rather than a 3Gb reference). The masking is required for the same reason – without it there is the possibility of non-full length reads mapping twice to the reference.

R2: The authors do not discuss if their selected gRNA sequences are polymorphic (eg common polymorphisms in GenBank or gnomAD).

gRNAs were designed to avoid the hypervariable region and most common polymorphic positions. It is possible that unknown polymorphic sites underlie gRNAs, however, the effect on the cutting efficiency depends on the position of the mismatch relative to the PAM site. Also, to minimize this risk at least 2 guides (or more, if required) are used per sample.

Exposition in the manuscript:

R2: The advances of this pipeline can be more clearly described to explain the actual experimental and computational pipeline. For example, the optimized protocol should be clearly explained (eg Exonuclease V only explained in supplement not in manuscript, and it's not clear where in the experimental pipeline this is required; similar for Proteinase K). The advances to optimize the protocol are important innovations, that are currently be lost/buried in the text.

The Cas9-mtDNA-enrichment experimental advances have been included in **Results** in **Preparing the Cas9-mtDNA-enrichment sequencing library** section and further clarified in Figure 1.

R2: Figure 1: it's unclear that multiple gRNAs are targeting the same sample prior to multiplexing.

Figure 1 has been updated with labels for aliquots to indicate the splitting of one sample.

R2: Figure 2: Mitogenome is non-standard term. Perhaps mtDNA or chrM?

Figure 2 has been corrected.

R2: Figure 2 and text: "Frequencies of alleles" is unclear. Often "allele frequency" indicates population allele frequency (eg MAF =minor allele frequency across individuals). I think you're talking about Variant Allele Fraction (sometimes termed VAF), which is the fraction $ALT_reads/(ALT_reads+REF_reads)$.

Changes have been made in the main text and Suppl. Materials.

R2: Table 1: it is unclear what that clinical sequencing lab's heteroplasmy was. Is this the "IrrPCR Illumina" columns? Or did you do your own IrrPCR and sequencing?

In Table 1 we improved the column heading and replaced "IrrPCR Illumina sequencing" by "Clinical laboratory results".

Reviewer #3 (Remarks to the Author):

R3: The authors used for their experiments the new ONT R10.4 flow cells that help to reduce the sequencing error but reported in their discussion (l. 338ff) that they obtained very similar results when using the older, still very common ONT R9.4 flow cells that have a higher sequencing error rate (Supp Table 5 & 6). However, the results for the ONT R9.4 flow cell differ very much between the two tables. In Supp Table 5, the fraction of chrM reads was just 23% and as low as the values that were obtained for the ONT R10.4 flow cells without enrichment. In Supp Table 6, the fraction of chrM reads is suddenly 60%. The same goes for the average divergence, which drops from 1.8% in Supp Table 5 to 1.1% in Supp Table 6. What's the difference between these results presented in these tables? Furthermore, neither of these tables is referenced in the main text, just in the Supplementary Material. The authors should add a reference to these tables in the discussion.

We apologize for the confusion on which flow cells were used in combination with which sequencing kits.

Supp Table 5 contains also a flow cell R9.4.1 paired with the SQK-LSK110 ("Kit10"), while in Supp Table 6 we used with all the flow cell types (R10.3, R10.4 and R9.4.1) exclusively the Q20+ Ligation Sequencing Kit ("Kit 12"). This explains the differences in Avg. divergence among Supp. Tables 5 and 6. The comparable levels of sensitivity and precision between the old Kit10/R9.4.1 and the new Kit12/R10.4 were due to the increased coverage obtained from Kit10/R9.4.1 sequencing.

We also improved the caption of the Supp. Table 5.

We made the statement clearer in the Discussion.

The tables have been referenced in the Discussion.

R3: Their new program baldur is able to phase SNVs over longer regions than it is typically possible using only Illumina short-read data (l. 232ff.). While the authors show that this works very well for a cell line sample and high frequency heteroplasmies (Supp Figure 4), they do not mention how they consider divergent sequences caused by sequencing errors or differences in sequencing length as they are common for clinical samples. It would be interesting to know in which way these type of uncertainties of the actual mtDNA haplotype are considered during phasing.

For SNV phasing, baldur does not attempt to reconstruct the entire haplotype containing the selected variants, but only considers the selected variants themselves. In this way the phasing of widely spaced variants can be observed without having to take into account sequence variabilities (either real or due to sequencing errors) in the intervening bases. A sentence has been added to the main text to make this clear.

R3: When studying a clinical sample with a large deletion in the mtDNA (AW6506) that could not be resolved using short-read sequencing, the authors used three guide RNA pairs to prepare the DNA for sequencing. Depending on the location of the cut sites, they were able to detect all different haplotypes (wildtype plus two deletions; Figure 3 c-e). This highlights the effect of the positioning of the cut sites for the success of recovering all possible haplotypes in a sample. While the authors stated that there was not correlation between the location of the cut sites and obtained coverage along the mtDNA genome (l. 321ff.), I couldn't find any data that could actually show this. For the cell line results, each cell line had their specific cut site and with the exception of the sample AW6506, there were no replicates of the same samples with different gRNAs. It would be interesting to see if the authors could actual show this experimentally by applying all four pairs gRNAs to the same cell line and comparing the resulting coverage and variance of the coverage across the mtDNA genome.

We apologize for the ambiguity. As stated in supplementary material, there are differences between guide efficiencies due to RNA folding, stability, complex formation, available off-target sequences, or potential mismatches. Therefore, we expected to see variation in coverage in between different guides.

In Figure 3 we show the circos plots with the full length reads only (“both”), as this graphically shows the proportions of the three mtDNA species in this sample. Clearly this demonstrated the superiority over methods applying amplification by IrPCR. Key is that with our Cas9-mtDNA enrichment method applied to this sample we carry out 3 reactions with three different guides.

In Figure 3c the gRNA mt3 cut site is present in all of the 3 mtDNA species, within this reaction aliquot the ratio of the three mtDNA species is not influenced by the efficiency of the guide. In Figure 3d the cut is outside the large deletion mtDNA species and thus only cuts the smaller deletion and wildtype species. The results from the three reaction aliquots are very consistent which is a nice internal control.

R3: As a minor comment, in Supp. Table 8, the columns "Heteroplasmic_fraction" and "Heteroplasmy_%" are identical and therefore redundant; furthermore, I don't think the label "Heteroplasmic_fraction" is the correct term of what is measured because a Heteroplasmic fraction

This has been corrected and highlighted.

REVIEWERS' COMMENTS

Reviewer #1 (Remarks to the Author):

Thank you for addressing my questions comprehensively. I have no further comments to add.

Reviewer #2 (Remarks to the Author):

My concerns have been addressed. A couple very minor issues:

- supplementary materials should be re-ordered
- VAF is variant allele fraction (not variant allele frequency)

Reviewer #3 (Remarks to the Author):

The authors have thoroughly addressed the concerns that I raised for the previous version of the manuscript and have adapted the manuscript accordingly. Therefore, I can recommend this version of the manuscript for publication and am looking very much forward to seeing it published.

REVIEWERS' COMMENTS

Reviewer #1 (Remarks to the Author):

Thank you for addressing my questions comprehensively. I have no further comments to add.

Reviewer #2 (Remarks to the Author):

My concerns have been addressed. A couple very minor issues:

- supplementary materials should be re-ordered
- VAF is variant allele fraction (not variant allele frequency)

The supplementary material was re-ordered.

VAF has been changed to “variant allele fraction”

Reviewer #3 (Remarks to the Author):

The authors have thoroughly addressed the concerns that I raised for the previous version of the manuscript and have adapted the manuscript accordingly. Therefore, I can recommend this version of the manuscript for publication and am looking very much forward to seeing it published.